# Synthesis and Antiproliferative Effect of Halogenated Coumarin Derivatives

**DOI:** 10.3390/molecules27248897

**Published:** 2022-12-14

**Authors:** Tinuccia Dettori, Giuseppina Sanna, Andrea Cocco, Gabriele Serreli, Monica Deiana, Vanessa Palmas, Valentina Onnis, Luca Pilia, Nicola Melis, Davide Moi, Paola Caria, Francesco Secci

**Affiliations:** 1Department of Biomedical Sciences, University of Cagliari, University Campus, 09042 Monserrato, CA, Italy; 2Department of Chemical and Geological Sciences, University of Cagliari, University Campus, 09042 Monserrato, CA, Italy; 3Department of Life and Environmental Sciences, University of Cagliari, University Campus, 09042 Monserrato, CA, Italy; 4Department of Mechanical, Chemical and Material Engineering, University of Cagliari, 09123 Cagliari, CA, Italy

**Keywords:** coumarins, antiproliferative activity, TPC-1 cells, apoptosis, ROS

## Abstract

A series of 6- and 6,8-halocoumarin derivatives have been investigated as potential antiproliferative compounds against a panel of tumor and normal cell lines. Cytotoxic effects were determined by the MTT method. To investigate the potential molecular mechanism involved in the cytotoxic effect, apoptosis assay, cell cycle analysis, reactive oxygen species (ROS), and reduced glutathione analysis were performed. Among the screened compounds, coumarins 6,8-dibromo-2-oxo-2*H*-chromene-3-carbonitrile **2h** and 6,8-diiodo-2-oxo-2*H*-chromene-3-carbonitrile **2k** exhibited the most antiproliferative effect in thyroid cancer-derived cells TPC-1. The apoptosis assay showed that both **2h** and **2k** induced apoptosis in TPC-1 thyroid cancer cells. According to these experiments, both coumarins induced a slight increase in TPC-1 cells in the G2/M phase and a decrease in the S phase. A significant increase in ROS levels was observed in TPC-1 treated with diiodocoumarin **2k**, while the dibromocoumarin **2h** induced a decrease in ROS in a dose and time-dependent manner.

## 1. Introduction 

2*H*-Chromen-2-one derivatives, mainly known as coumarins, represent a large class of heterocyclic compounds in which the core structure is constituted by a 6-membered α-pyrone ring fused with an aromatic benzene ring. Coumarins have been isolated from a wide number of plants, mainly identified in roots, fruits, and leaves, and isolated from their extracts. As well, several coumarin derivatives have been extracted from various microorganisms, marine algae, and fungi [1,2,3,4]. A part of these natural sources, a certain number of synthetic organic methods developed over the decades since 1868 [5,6] allow straightforward access to a large number of 2*H*-chromenone scaffolds starting from low-cost and commercially available compounds [7,8]. The coumarin scaffold represents a privileged architecture in the design of fluorescent probes for bioimaging and sensing applications [9,10,11,12], but in particular, it constitutes a key intermediate that has acquired a special place in heterocyclic and medical chemistry research. These compounds are of fundamental importance to accessing new molecular architectures and provide the opportunity to extend the structural diversity and chemical space exploration in drug discovery [13]. Investigations on these derivatives highlighted a wide spectrum of biological properties, including anticoagulant, anticancer, anti-inflammatory, antioxidant, antiviral, antimicrobial, and other activities [14,15,16,17]. Moreover, coumarin derivatives received particular attention due to their photochemotherapy and potential therapeutic employments on several human tumors [18,19,20,21,22,23,24,25,26,27,28]. Indeed, coumarins were investigated for their ability to arrest cancer cell proliferation by blocking the cell cycle in G0/G1 [29] and G2/M phases [30] and/or generating free radical species with pro-apoptotic effects [31]. Furthermore, the presence and position of an electron-withdrawing group in the aromatic ring of the coumarin may enhance its biological activity. The best pharmaceutical results can be expected by introducing a halogen atom, which improved the lipophilic properties of the molecules with an increase in the penetration of lipid membranes, as confirmed by several antibiotics, antineoplastic agents, CNS depressants, CNS stimulants, and antipsychotics [32]. Halogenated coumarin derivatives have also been reported for their anti-malarial, protein kinase, and MAO-B inhibitor activity [33,34,35]. Furthermore, halogen bonding has been shown to increase the affinity of potential anticancer drugs [36]. Due to their attractive pharmacological properties, we synthesized and investigated a series of 6- and 6,8-halogenated coumarins, evaluating their potential antiproliferative activity against a panel of tumor cell lines and their molecular mechanism involved in the cytotoxic effects.

## 2. Chemistry 

Synthesis of halocoumarins **2a**–**k** [37,38,39,40] was accomplished using the procedure described in Figure 1. The condensation of halosalicyladehydes **1a**–**e** with carbonyl compounds such as diethyl malonate, methyl 2-(methylsulfonyl)acetate, and malononitrile were performed in the presence of a selection of additives accessing compounds **2a**–**k** in good to excellent yields [11]. On the other hand, the coumarin **2j** was obtained by acid hydrolysis of the corresponding ethyl ester **2i** [37], as summarized in Figure 1.

## 3. Results and Discussion

### 3.1. Antiproliferative Activity of Coumarins ***2a**–**k*** against a Panel of Cancer and Normal Cell Lines

The cytotoxic effect of coumarins **2a**–**k** was tested on several cancer and normal cell lines: lung cancer cell line (SK-MES-1), thyroid cancer cell lines (K-1; B-CPAP ant TPC-1), lung fibroblast (MRC-5) and thyroid normal cell line (Nthy-ori-3-1) by MTT (3-[4,5-dimethylthiazol-2-yl]-2,5 diphenyltetrazolium bromide) assay [41,42]. Several studies demonstrated camptothecin activity in tumors of various origins, so we decided to employ this reference drug as internal control in our in vitro assays [43]. The CC_50_ values, calculated as the concentrations inhibiting the cells growth by 50%, are reported in Table 1.

None of the coumarins showed cytotoxic activity on the normal cell lines MRC-5 and Nthy-ori-3-1, while coumarin **2k** showed moderate antiproliferative activity against thyroid cancer cells (CC_50_ = 57 µM in K-1, CC_50_ = 44 µM in TPC-1, CC_50_ = 46 µM in B-CPAP). Furthermore, the coumarin **2h** showed antiproliferative activity against K-1 and TPC-1 cells (CC_50_ = 60 µM and 90 µM, respectively). These results suggested that thyroid cancer cells may be more sensitive cell lines to compounds **2k** and **2h** with respect to viability. 

### 3.2. Apoptosis Induction of Coumarins ***2k*** and ***2h***

To determine whether the two studied coumarins **2k** and **2h**, induce cell death through apoptosis or necrosis, we performed double staining with fluorescent Annexin V-FITC/propidium iodide (PI) apoptosis detection kit in TPC-1 and Nthy-ori-3-1 as control cells. This assay is based on the ability of Annexin to recognize phosphatidylserine (PS), a membrane phospholipid exposed on the cell surface of apoptotic cells [44], while PI is able to bind the DNA of necrotic cells. Thanks to this combination, three cell populations are obtained: cells negative for Annexin V-FITC and PI indicate live cells, Annexin V-FITC-positive/PI-negative and Annexin V-FITC-positive/PI-positive indicate early and late apoptotic cells, while cells negative for Annexin V-FITC and PI-positive are necrotic. As reported in Figure 1A, the TPC-1 cells showed a significant decrease in the number of viable cells after 24 h of treatment with **2k** (at 20 μM concentration) when compared to untreated cells (87% vs. 62%). The percentage of total apoptotic cells resulted in 34% in **2k**-treated cells vs. 10% untreated cells (*p* < 0.05). 

Compound **2h** induces a decrease in cell viability if compared with untreated cells (77.7% vs. 87%) and 18% of total apoptotic cells, while **2k** was able to induce a significant apoptotic activity against cancer cells compared with **2h** analog (*p* < 0.05). No significant induction of apoptosis was observed in Nthy-ori-3-1 normal thyroid cells (Figure 1B).

### 3.3. Effect of Coumarins ***2k*** and ***2h*** on the Different Phases of the Cell Cycle

Among the potential mechanisms involved in the treatment of cancer, there is the arrest of the cell cycle [45]. To detect whether the cytotoxicity of coumarins **2k** and **2h** resulted from cell cycle progression, the effects on the cell cycle of TPC-1 cells were analyzed by flow cytometry after labeling with PI (propidium iodide)/RNase Staining Buffer detection kit. As shown in Figure 2, treatment with **2k** induces an increase in TPC-1 cells in the G2/M phase compared with the untreated cells (*p* < 0.01) and simultaneously decreases the cell population in the S phase (Figure 2A). The same trend was observed in **2h**-treated cancer cells. No effect on the cell cycle was observed in control cells (Figure 2B). According to apoptosis results, the cell cycle arrest could be responsible for the cytotoxicity in cancer cells. In the present study, it was observed that coumarin **2k** determined cell arrest of TPC-1 cells in the G2/M phase, indicating the apoptotic mode of cell death and inhibition of DNA synthesis (S phase reduction). This finding is consistent with previous investigations reporting that coumarins able to induce cytotoxicity by inducing G2/M and S phases arrest in different cancer cell lines [46,47].

### 3.4. Effect of Coumarins ***2k*** and ***2h*** on ROS Production and Their Redox Potential

It has been reported that coumarins and their derivatives are able to regulate the reactive oxygen species (ROS) [48]. For example, esculetin induces cytotoxicity in pancreatic cancer cells decreasing ROS and protein levels of the ROS-dependent transcription factor NF-κB [49]. On the other end, coumarins induce the generation and accumulation of ROS in cancer cells causing cell cycle arrest and apoptotic cell death [50]. In our study, TPC-1 cells were exposed to coumarin **2k** (at 20 and 100 μM concentrations) for different time intervals in which a significant increase in ROS generation started as early as 30 min, and the relative fluorescence continually grew until 120 min (*p* < 0.001). In contrast, treatment with coumarin **2h** induced a subsequently decreased ROS generation using both concentrations (Figure 3A). A similar effect, pro-oxidant of coumarin **2k** and antioxidant of coumarin **2h** was observed in Nthy-ori-3-1 cells but only with 100 μM, the higher concentration used (Figure 3B). These results suggest that ROS level significantly increased in **2k**-treated TPC-1 cells, causing a reduction in cancer cell growth and subsequent induction of apoptosis and cell cycle arrest. Conversely, **2h** produces an antiproliferative response in TPC-1 cells by scavenging ROS. In an attempt of a better understanding of the results, such compounds were explored from an electrochemical point of view by means of cyclic voltammetry in anhydrous acetonitrile. As reported in Appendix A, the dibromo derivative **2h** shows an oxidation peak at a less positive potential with respect to the diiodo derivative **2k** (0.6 V vs. 0.75 V). On the other hand, **2k** shows a reduction feature at less negative potential compared to **2h** (−0.95 V vs. −1.02 V). Although the biological process under these opposite effects needs more investigation, these results reflect an easier oxidation of **2h** over **2k** and an easier reduction of **2k** over **2h**, which are consistent with the observed experimental behavior. 

### 3.5. Modulation of Intracellular Content of Reduced and Oxidized Glutathione (GSH/GSSG)

Redox perturbation was further investigated by measuring intracellular antioxidant species. In Nthy-ori3-1 cells, coumarin **2k** at 20 µM improved the GSH concentration measured as GSH/GSSG ratio (*p* < 0.001), leading to values close to 200% with respect to the untreated cells (100%), while coumarin **2h** increased the level of GSH but with less evidence (about 40% more than the untreated control cells, *p* < 0.01) (Figure 4). Interestingly, an opposite action occurred in TPC-1 tumor cells for coumarin **2h**, whereas coumarin **2k** increased the availability of reduced GSH with values like those found in Nthy-3ori-1 cells (about 200%, *p* < 0.01 vs. control), but coumarin **2h** also led to a huge increase in GSH reduced up to 500% (*p* < 0.001 vs. control and **2k**). It can therefore be noted that coumarin **2h**, which has been shown to have antioxidant behavior, has performed its function in a striking way precisely in TPC-1 where oxidative stress is greater, and GSH is then produced in considerable quantities for the maintenance of cellular functions and homeostasis. In this case, we speculate that there could be occurred a cell response to a stimulus induced by coumarin **2h** in addition to that lead by the oxidative environment and that the antioxidant effects that were observed in the experiments shown before may be related in part or totally to this process. The Keap1/Nrf2/ARE system is a mechanism that plays a key role in the pathogenesis and progression of several diseases. It is a central defensive mechanism against oxidative stress.

Some coumarins have shown the ability to activate Nrf2 signaling in different cell lines and animal models [51]. A potential mechanism of action of our coumarins, especially the **2h,** which has shown marked antioxidant behavior, may consist of the activation of the Keap1/Nrf2/ARE signaling pathway, which has recently been reported to be modulated by different coumarins of origin natural [52] and is known to be involved in the GSH biosynthesis [53]. However, further studies should be conducted to well address this important question.

There are certainly two aspects to be taken into consideration that may have influenced the results of this investigation: the first is the difference in initial concentration of GSH in the two cell types since in TPC-1 it has already been shown in previous studies GSH levels are higher than normal cells [54] and was also ascertained in the present investigation (data not shown). This higher concentration can affect the biochemical processes that lead to the depletion of GSH or its formation starting from the GSSG, which undergoes the reducing and/or oxidative action of the tested coumarins; the second aspect is linked to the simultaneous use of cysteine (Cys) by the cells as an antioxidant supply, both as such and as a precursor to reconstitute GSH [41]. Cys can react with the substances tested or be depleted first with respect to GSH in relation to the oxidative stimulus that occurs in the cancer cellular environment.

## 4. Experimental Section

### 4.1. Chemistry

#### 4.1.1. General Remarks

Unless stated otherwise, respectively, the synthesis of compounds **2a**–**k** was performed at the indicated temperature in a round bottom flask equipped with a stirring bar and a condenser if needed. Commercially available reagents were used as received unless otherwise noted. The salicylaldehydes 1 used in this work were purchased from Sigma Aldrich, TCI of Fluorochem. ^1^H NMR spectra were recorded on a Bruker Avance III HD 600 spectrometer (Bruker, Bremen, Germany) at 300.15 K using CDCl_3_ (ref. 7.27 ppm) as a solvent. ^13^C NMR was recorded on a Bruker Advance III HD 600 (Bruker, Bremen, Germany) at 126 MHz (ref. CDCl_3_ 77.00 ppm) at 300.15 K using CDCl_3_ as solvent. Chemical shifts (δ) are given in ppm. Coupling constant values (J) are reported in Hz. Infrared spectra were recorded on an FT-IR Bruker Equinox-55 spectrophotometer (Bruker, Bremen, Germany) and were reported in wavenumbers (cm^−1^). Low mass spectra analyses were recorded on an Agilent-HP GC-MS (E.I. 70 eV). High-resolution mass spectra (HRMS) were obtained using a Bruker High-Resolution Mass Spectrometer (Bruker, Bremen, Germany) in fast atom bombardment (FAB^+^) ionization mode. Melting points were determined with a Büchi M-560 Analytical (Buchi, Flawil, Swiss). Thin layer chromatography was performed using 0.25 mm Aldrich silica gel 60-F plates. Flash chromatography was performed using Merk 70–200 mesh silica gel. Yields refer to chromatography and spectroscopically pure materials.

#### 4.1.2. General Procedure for the Synthesis of Ethyl 2-oxo-2H-chromene-3-carboxylate derivatives **2b**, **2d**, **2g**, and **2i**

Piperidine (0.17 g, 0.002 mol) was added to a solution of halosubstituted 2-hydroxybenzaldehyde (0.02 mol) and diethyl malonate (3.2 g, 0.02 mol) in EtOH (100 mL) and some drops of AcOH. The mixture was stirred at 50 °C until reaction completion (16 h). The reaction mixture was concentrated under vacuum, and the residue was crystallized by MeOH to obtain the resulting title compounds as crystalline solids. 

##### Ethyl 6-fluoro-2-oxo-2H-chromene-3-carboxylate (2b) 

Yield 76%. IR (KBr): ν = 3069, 1748 cm^−1^. P.f. 161–164 °C ^1^H NMR (600 MHz, CDCl_3_): δ = 8.42 (s, 1H), 7.61–7.54 (m, 2H), 7.31 (d, *J* = 8.5 Hz, 1H), 4.41 (q, *J* = 7.1 Hz, 2H), 1.40 (t, *J* = 7.1 Hz, 3H). ^13^C NMR (151 MHz, CDCl_3_): δ = 162.81, 156.15, 153.63, 147.22, 134.27, 130.29, 128.57, 119.71, 118.97, 118.41, 62.35, 14.33. All analytical data were in suitable accordance with the reported data [55].

##### Ethyl 6-bromo-2-oxo-2H-chromene-3-carboxylate (2d)

Yield 90%. IR (KBr): ν = 3006, 1734 cm^−1^. P.f. 184–186 °C. ^1^H NMR (600 MHz, CDCl_3_): δ = 8.44 (s, 1H), 7.76 (d, *J* = 2.3 Hz, 1H), 7.74 (dd, *J* = 8.8, 2.3 Hz, 1H), 7.28 (d, *J* = 4.9 Hz, 1H), 4.44 (q, *J* = 7.1 Hz, 2H), 1.43 (t, *J* = 7.1 Hz, 3H). ^13^C NMR (151 MHz, CDCl_3_): δ = 162.77, 156.07, 154.08, 147.12, 137.05, 131.65, 119.65, 119.48, 118.66, 117.47, 62.34, 14.32. HRMS-ESI: calcd for C_12_H_9_BrO_4_ (M-Na 318.9582), found (M-Na, 318.9581). All analytical data were in suitable accordance with the reported data [56].

##### Ethyl 6,8-dibromo-2-oxo-2H-chromene-3-carboxylate (2g) 

Yield 87%. IR (KBr): ν = 3030, 1776 cm^−1^. P.f. 231–233 °C. ^1^H NMR (600 MHz, CDCl_3_): δ = 8.37 (s, 1H), 7.95 (d, *J* = 2.2 Hz, 1H), 7.69 (d, *J* = 2.2 Hz, 1H), 4.40 (q, *J* = 7.1 Hz, 2H), 1.39 (t, *J* = 7.1 Hz, 3H). ^13^C NMR (151 MHz, CDCl_3_): δ = 162.32, 154.94, 151.05, 146.68, 139.53, 130.85, 120.31, 120.16, 117.31, 111.43, 62.48, 14.27. HRMS-ESI: calcd for C_12_H_8_Br_2_O_4_ (M-Na 396.8687), found (M-Na, 396.8688). All analytical data were in suitable accordance with the reported data [55].

##### Ethyl 6,8-diiodo-2-oxo-2H-chromene-3-carboxylate (2i)

Yield 73%. IR (KBr): ν = 3060, 1736 cm^−1^. P.f. 199–201 °C. ^1^H NMR (600 MHz, CDCl_3_): δ = 8.33 (d, *J* = 1.8 Hz, 1H), 8.30 (s, 1H), 7.88 (d, *J* = 1.8 Hz, 1H), 4.41 (q, *J* = 7.1 Hz, 2H), 1.39 (t, *J* = 7.1 Hz, 3H). ^13^C NMR (151 MHz, CDCl_3_): δ = 162.36, 155.21, 154.32, 150.80, 146.63, 137.90, 120.11, 120.09, 88.00, 85.21, 62.45, 14.29. All analytical data were in suitable accordance with the reported data [55].

#### 4.1.3. General Procedure for the Synthesis of 2-oxo-2H-chromene-3-carbonitrile Derivatives **2a**, **2c**, **2e**, **2h** and **2k**

Malononitrile (0.05 g, 0.8 mmol) was added to a solution of the substituted salicylaldehyde (0.4 mmol) in aqueous 0.05 M Na_2_CO_3_ (1.5 mL), and the mixture was stirred at room temperature. Within 5 min, a yellow solid started to precipitate from the reaction mixture, which was kept stirring at room temperature for 24 h. Then, 37% aqueous HCl (3.7 equiv.) was added to the suspension. The reaction mixture was heated at 80–90 °C for 6 h. The suspension was cooled to room temperature in an ice bath, and the solid was filtered and washed with water and diethyl ether leading to the pure title products.

##### 6-Fluoro-2-oxo-2H-chromene-3-carbonitrile (2a)

Yield 71%. IR (KBr): ν = 3077, 1723 cm^−1^. P.f. 167–169 °C. ^1^H NMR (600 MHz, CDCl_3_): δ = 8.23 (s, 1H), 7.48–7.38 (m, 2H), 7.30 (dd, *J* = 7.3, 2.8 Hz, 1H). ^13^C NMR (151 MHz, CDCl_3_): δ = 160.10, 158.45, 156.05, 150.94, 150.84, 150.82, 123.29, 123.13, 119.41, 119.36, 117.84, 114.56, 114.40, 113.27, 104.91. All analytical data were in suitable accordance with the reported data [57].

##### 6-Chloro-2-oxo-2H-chromene-3-carbonitrile (2c)

Yield 69%. IR (KBr): ν = 3043, 1732 cm^−1^. P.f. 171–174 °C. ^1^H NMR (600 MHz, CDCl_3_): δ = 8.20 (s, 1H), 7.66 (dd, *J* = 8.9, 2.4 Hz, 1H), 7.60 (d, *J* = 2.4 Hz, 1H), 7.37 (d, *J* = 8.9 Hz, 1H). ^13^C NMR (151 MHz, CDCl_3_): δ = 155.85, 153.06, 150.59, 135.49, 131.38, 128.38, 119.06, 118.13, 113.22, 104.84. All analytical data were in suitable accordance with the reported data [57].

##### 6-Bromo-2-oxo-2H-chromene-3-carbonitrile (2e)

Yield 75%. IR (KBr): ν = 3085, 1732 cm^−1^. P.f. 167–172 °C. ^1^H NMR (600 MHz, CDCl_3_): δ = 8.19 (s, 1H), 7.80 (dd, *J* = 8.9, 2.3 Hz, 1H), 7.74 (d, *J* = 2.3 Hz, 1H), 7.31 (d, *J* = 8.8 Hz, 1H). ^13^C NMR (151 MHz, CDCl_3_): δ = 155.78, 153.53, 150.47, 138.30, 131.44, 119.29, 118.54, 118.61, 113.19, 104.80. All analytical data were in suitable accordance with the reported data [11].

##### Ethyl 6,8-dibromo-2-oxo-2H-chromene-3-carboxylate (2h)

Yield 87%. IR (KBr): ν = 3030, 1776 cm^−1^. P.f. 231–233 °C. ^1^H NMR (600 MHz, CDCl_3_): δ = 8.18 (s, 1H), 8.04 (d, *J* = 2.1 Hz, 1H), 7.71 (d, *J* = 2.1 Hz, 1H). ^13^C NMR (151 MHz, CDCl_3_): δ = 154.92, 150.50, 150.29, 140.88, 130.68, 119.34, 118.46, 112.79, 112.23, 105.42. HRMS-ESI: calcd for C_12_H_8_Br_2_O_4_ (M-Na 396.8687), found (M-Na, 396.8689). All analytical data were in suitable accordance with the reported data [11].

##### 6,8-diiodo-2-oxo-2H-chromene-3-carbonitrile (2k)

Yield 68%. IR (KBr): ν = 3058, 1721 cm^−1^. P.f. 221–227 °C. ^1^H NMR (600 MHz, DMSO): δ = 8.38 (s, 1H), 8.36 (d, *J* = 1.9 Hz, 1H), 8.22 (d, *J* = 1.9 Hz, 1H). ^13^C NMR (151 MHz, DMSO): δ = 163.90, 156.36, 153.21, 148.42, 144.32, 137.70, 122.82, 120.93, 89.16, 86.19. HRMS-ESI: calcd for C_10_H_3_I_2_NO_2_ (422.8253), found (M + H^+^), 423.8301. 

#### 4.1.4. 6-Bromo-3-(methylsulfonyl)-2H-chromen-2-one (**2f**)

Piperidine (0.17 g, 0.002 mol) was added to a solution of 4-bromo-2-hydroxybenzaldehyde (4 g, 0.02 mol) and ethyl 2-(methylsulfonyl)acetate (3.32 g, 0.02 mol) in EtOH (100 mL) and some drops of AcOH. The mixture was brought to 50 °C and kept under stirring until reaction completion (16 h). The reaction mixture was concentrated under vacuum and the residue crystallized in hot MeOH to obtain a white crystalline solid. Yield 75%. IR (KBr): ν = 3052, 1736 cm^−1^. P.f. 164–169 °C. ^1^H NMR (600 MHz, CDCl_3_): δ = 8.56 (s, 1H), 7.85–7.79 (m, 2H), 7.33 (d, *J* = 8.7 Hz, 1H), 3.34 (s, 3H). ^13^C NMR (151 MHz, CDCl_3_): δ = 155.51, 154.21, 146.35, 138.33, 132.50, 128.93, 119.01, 118.67, 118.42, 41.90. HRMS-ESI: calcd for C_10_H_7_BrNaO_4_S (324.9146), found (M-Na, 324.9147). All analytical data were in suitable accordance with the reported data [11].

#### 4.1.5. 6,8-diiodo-2-oxo-2H-chromene-3-carboxylic Acid (**2j**) 

To a solution of ethyl 6,8-diiodo-2-oxo-2*H*-chromene-3-carboxylate (**2i**) (0.01 mol, 4.7 g) in H_2_O/MeOH (1:5 *v*/*v*; 10 mL), LiOH (0.03 mol, 0.72 g) was added, and the solution was stirred overnight at room temperature. Then, HCl 3 N (10 mL) was added, and the formed was filtered off, washed with water, and dried under vacuum. Yield 81%. IR (KBr): ν = 3064, 1719 cm^−1^. P.f. 289–291 °C. ^1^H NMR (600 MHz, CDCl_3_): δ = 8.42 (d, *J* = 1.9 Hz, 1H), 8.07 (s, 1H), 7.87 (d, *J* = 1.9 Hz, 1H). ^13^C NMR (151 MHz, CDCl_3_): δ = 155.14, 153.80, 152.21, 150.04, 137.61, 119.25, 112.78, 105.27, 89.11, 85.88. All analytical data were in suitable accordance with the reported data [31].

#### 4.1.6. Electrochemical Measurements 

The electrochemical properties of compounds **2h** and **2k** were investigated by cyclic voltammetry (CV) technique under an inert atmosphere. The measurements have been carried out at 100 mV s^−1^ with a three-electrode cell using a platinum tip as a working electrode, a platinized titanium net as a counter electrode, and a platinum wire as a pseudo-reference electrode. The electrochemical measurements were performed at room temperature using an AUTOLAB PGSTAT302N (Metrohm, Herisau, Switzerland) potentiostat/galvanostat controlled with the NOVA software. All the solutions were prepared in anhydrous degassed acetonitrile with 0.1 M [*n-*Bu_4_N][PF_6_] as supporting electrolyte: in this condition, the Fc^+/0^ redox couple is centered at +0.45 V.

### 4.2. Biological Evaluations

#### 4.2.1. Cell Lines and Cell Culture 

The papillary thyroid carcinoma (PTC)-derived cell lines (TPC-1 and B-CPAP) were kindly provided by Dr. Fusco (Medical School, University Federico II of Naples, Naples, Italy). K-1 PTC-derived cell line and non-tumorigenic thyroid Nthy-ori-3-1 cell line were purchased from Health Protection Agency Culture Collections. The PTC-derived cell lines were maintained in DMEM/F12 (Gibco-BRL. Life Technologies, Milan, Italy)] supplemented with 10% fetal bovine serum (Gibco-BRL) at 37 °C in humidified 5% CO_2_. SK-MES-1 and MRC-5 were purchased from American Type Culture Collection (ATCC) and were maintained in Mem-E medium, supplemented with 10% fetal bovine serum, 100 units/mL penicillin G, and 100 μg/mL streptomycin (Gibco-BRL) at 37 °C in humidified 5% CO_2_. 

#### 4.2.2. Cell Viability Assays 

The effect of test compounds on the viability of cells was detected by MTT assay. The thyroid cells (7.5 × 10^3^ cells/mL), SK-MES-1, and MRC-5 (8 × 10^4^ cells/mL) were seeded into a 96-well plate and allowed to adhere for 16 h to culture plates before the addition of the compounds. 

Then, compounds were added at different concentrations (0.8–100 μM), and cells were further incubated for 24 h. Cells with no treatment were used as a negative control. After incubation, cells were treated with MTT solution for 3 h at 37  °C. Then, the MTT solution was removed, and 150 µL of DMSO was added to the wells. The absorbances were measured at 570 nm using a TECAN microplate reader (Infinite 200, Tecan, Salzburg, Austria). Viability data were reported as % of control (untreated cells) for each cell line.

#### 4.2.3. Cell Cycle Analysis by Flow Cytometry

The cells were seeded (5 × 10^4^ cells/mL) in six-well plates (Corning, Tewksbury, MA, USA). After 24 h, they were treated with coumarins (20 µM) for 24 h. Cell cycle analysis was performed by using the Invitrogen™ FxCycle™ PI/RNase Staining Solution (according to the manufacturer’s instructions, Thermo Fisher Scientific, Waltham, MA, USA), and then the DNA content was evaluated by flow cytometry (MoFloAstrios EQ, Beckman Coulter, Brea, CA, USA). The analysis of cell cycle phase distribution was accomplished via Kaluza Flow Cytometry Analysis Software (Software Version 1.2, Beckman Coulter) by setting 3 gates in each single parameter histogram: G0/G1, S, and G2/M.

#### 4.2.4. Cell Apoptosis Assay 

To investigate cell death induced by coumarins treatment, a flow cytometric analysis using the cell apoptosis kit Annexin V/propidium iodide (PI) double staining uptake (Life Technologies, Monza, Italy) was employed. Control and PTC-derived cells, at the density of 5 × 10^4^ cells/mL, were seeded in 6-well plates (Corning) with complete DMEM/F12. First, the cells were treated with 20 µM of coumarins for 24 h. Cells were washed once with PBS 1X and stained according to the kit’s protocol. Stained cells were then analyzed by flow cytometry, measuring the fluorescence emission at 530 and 620 nm using a 488 nm excitation laser (MoFloAstrios EQ, Beckman Coulter). Cell apoptosis was analyzed using Software Summit Version 6.3.1.1, Beckman Coulter. 

#### 4.2.5. Determination of Intracellular ROS Production

PTC-derived and control cells were seeded in 96-well plates (7.5 × 10^3^/mL) to detect intracellular ROS production and grown for 24 h. Cells were then washed with PBS 1X solution and incubated for 30 min with 2′,7′-dichlorofluorescein diacetate probe (H_2_-DCF-DA) (Merck, Milan, Italy) (10 μM), as previously described [41]. H_2_-DCF-DA was then removed, and cells were treated with 20 μM of coumarins, the hydrogen peroxide (H_2_O_2_, 1 mM), as a positive control. After 1 h of incubation, excess H_2_-DCF-DA was removed and replaced with PBS, and then ROS levels were measured by using a microplate reader (Infinite 200, Tecan, Salzburg, Austria) at a controlled temperature of 37 °C. The measurement was performed using an excitation of 490 nm and an emission of 520 nm. ROS production was evaluated for 2 h and monitored, taking readings at intervals of 5 min.

#### 4.2.6. Determination of Intracellular Reduced and Oxidized Glutathione 

Reduced glutathione (GSH) and oxidized glutathione (GSSG) levels were determined with high-performance liquid chromatography coupled with an electrochemical detector (HPLC-ECD), as described in previous studies [58,59]. In detail, Nthy-ori3-1 and TPC-1 cells were seeded in 6-well plates at the density of 1 × 10^5^ cells/2 mL and incubated for 24 h. Cells were then treated with **2k** and **2h** (20 µM) for 24 h. After the incubation, cells were scraped and extracted with 150 µL of 10% meta-phosphoric acid and 150 µL of 0.05% trifluoroacetic acid (TFA) (Merck, Milan, Italy) solution. After centrifugation, 10 µL of supernatant was collected for protein determination, and the remaining part was injected into the HPLC system. GSH and GSSG amounts were measured using an HPLC (Agilent 1260 infinity, Agilent Technologies, Palo Alto, CA, USA) equipped with an electrochemical detector (DECADE II Antec, Leyden, The Netherlands) and an Agilent interface 35900E. A calibration curve was created using standards of GSH and GSSG (Merck, Milan, Italy), injected with different concentrations. Data were collected and expressed as a percentage of the control cells (100%) of the ratio between GSH and GSSG normalized using µg of total proteins.

#### 4.2.7. Statistical Analysis

All data are presented as means ± SD for three independent experiments. Statistical differences between the two groups were analyzed using a Student’s *t*-test by GraphPad Prism 5.0 (GraphPad Software, San Diego, CA, USA). *p* < 0.05 was regarded as significant.

## 5. Conclusions

In this study, a series of 6- and 6,8-halocoumarin derivatives was prepared and investigated for their antiproliferative activity. The 6,8-dibromo-2-oxo-2*H*-chromene-3-carbonitrile **2h** and 6,8-diiodo-2-oxo-2*H*-chromene-3-carbonitrile **2k** exhibited antiproliferative effect in thyroid cancer-derived cells TPC-1 and more importantly did not affect the growth of thyroid normal cell line Nthy-ori-3-1. The TPC-1 cells decreased viability produced by both coumarins **2h** and **2k** has been related to apoptosis induction as well as to a slight increase in the G2/M phase and a decrease in the S phase. The ROS level increase produced by the 6,8-diodocoumarin **2k** may contribute to the TPC-1 cells’ cytotoxic activity. Therefore, these dihalogenated coumarins are promising templates for the development of novel antiproliferative compounds.

## Data Availability

The data presented in this study are shown in this paper.

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
