# Peer review of "Synthesis and Antiproliferative Effect of Halogenated Coumarin Derivatives"

_molecules, 2022, doi:10.3390/molecules27248897_

Round 1

Reviewer 1 Report

The work has some advantages, but from my point of view the discovery is not attractive enough for publication in its current version in such a reputable journal as Molecules.

First of all, the work did not emphasize the novelty of the research and its potential significance. The conducted research has not been summarized - there is no Conclusions section. In addition, the authors presented the current state of knowledge on halogenated coumarin derivatives very generally and too briefly.

Most of the synthesized compounds are known. Only 2 out of 11 coumarins show some antiproliferative activity. Compared to camptothecin, these activities is negligible - why did the authors choose camptothecin as a reference drug?

Author Response

Dear reviewer, please check our attached file.

best regards,

F Secci

Reviewer 2 Report

The authors present a synthesis and antiproliferative activity of several halogenated coumarin derivatives. It is not clear, why halogenation should improve the cytotoxic effect. In any case, two compounds showed moderate cytotoxic activity in two of the investigated cancer cell lines. The mechanism of the action was investigated. The work will be suitable for publication in Molecules if the following issues have been resolved:

Minor issues:

- line 17: A series

- Schemes, Figures, etc. should be written with capital first letters throughout

- line 117: compound numbers in bold

- line 162: redox, not Redox

Major Issues:

- Yields should be provided in the synthetic schemes.

- Calculated values for HRMS analysis must be provided for the form that is observed in the measurement.

- Printouts of the NMR spectra must be provided in supplementary material

- Interaction with Keap1/Nrf2/ARE pathway is proposed, but no evidence is provided

- Conclusion part is completely missing

Author Response

Dear Reviewer, please check the attached file.

best regards, 

F Secci

Round 2

Reviewer 1 Report

The manuscript was revised according to the comments and suggestions of the Reviewers.

Reviewer 2 Report

The authors have fulfilled all the requests of the reviewer. Even though the activity of the molecules is rather low, the mechanistic investigation related to the molecular mode of action can be of interest. Therefore, in the reviewer's opinion, in its current form, the article might be suitable for publishing in Molecules.